# Economic Impact Analysis of Farmers' Markets in the Washington, DC Metropolitan Area: Evidence of a Circular Economy

**Kamran Zendehdel** [1], **Brian W. Sloboda** [2,*] **and Eric Chad Horner** [1]

1   College of Agricultural, Urban Sustainability, and Environmental Sciences University of District of Columbia, Washington, DC 20008, USA; kamran.zendehdel@udc.edu (K.Z.); eric.horner@udc.edu (E.C.H.)
2   Global Campus, School of Business, University of Maryland, Largo, MD 20774, USA
*   Correspondence: brian.sloboda@faculty.umgc.edu

**Abstract:** Consumer interest in farmers' markets (FMs) has dramatically increased during the past decade. The number of FMs in the United States has grown from 1755 in 1994 to 8140 in 2019 (USDA, 2019). To evaluate the economic impacts (EIs) of FMs in the Washington, DC metropolitan area, we collected FMs' consumer data and used IMPLAN-based social accounting matrices to evaluate the direct, indirect, and induced economic impacts of FMs. The empirical results from IMPLAN provide the direct gross sales, income figures, and an estimate of the number of jobs in the study region. The results show the average total output of USD 36,181,059, total employment of 663 people, total value-added creation of USD 19,019,226, and total labor income created of USD 8,653,350 in the region. The FM average income multiplier is 1.51, which indicates that a USD 1 increase in personal income (PI) for an FM translates into USD 1.51 in PI across the economy of the region. We also highlight the impact of FMs as an important component of the circular economy (CE). To this end, we present a qualitative approach examining the potential of a CE as applied to the farmers' markets in the Washington, DC metropolitan area using qualitative data from focus groups. The goal of the circular economy is to provide more sustainability in the local economy.

**Keywords:** farmers' markets; consumer data; economic impact analysis; IMPLAN-based social accounting matrices (SAMs); circular economy; urban sustainability

## 1. Introduction

Consumers in the past decade have increased their patronage of farmers' markets due to consumers' desire for fresh and locally produced food. Within the Washington, DC metropolitan area alone, farmers' markets have increased in number, with 154 farmers' markets in this region. Farmers' markets have existed since the 17th century in the United States, with the first known farmers' market in established Boston in 1634 by the English Colonial Governor John Winthrop [1].

Increased market participation in recent decades may have occurred for various social and economic reasons. The analysis in this paper measures the economic contribution of farmers' markets to the regional economy of the metropolitan Washington, DC area. Stated differently, with the growth in farmers' markets in recent years along with the increases in interest and information about the benefits of purchasing and consuming local, fresh food products, we examined how this rising interest provides economic benefits to the regional economy. Additionally, we aimed to determine if it improves sustainability in the local economy. We delved somewhat into the circular economy, which is tied to urban sustainability, using qualitative data obtained from focus groups conducted during the research. Numerous studies have addressed the economic impacts of farmers' markets on states and regions. However, no empirical work was completed for the Washington DC metropolitan area. This research has two objectives.

- Measure the economic contribution of farmers' markets using economic impact analysis to the economy of the Washington DC metropolitan area;
- Develop a preliminary assessment of the application of the concepts of the circular economy as it pertains to farmers' markets in the Washington DC metropolitan area.

In this paper, we also provide an assessment of the existing literature concerning the economic impacts and methods used to assess the economic impacts of farmers' markets on a regional or state economy. We investigate the circular economy impacts of FMs and how farmers' markets may possibly tie into sustainability. We suspect there is a circular economy in food production in farmers' markets in the Washington DC metropolitan area, but we do not have a reliable data source to identify it. Consequently, we resorted to the focus groups which could identify a potential circular economy in food production. We delve more on the focus groups later in this paper. In Section 2, we discuss the data that supplement IMPLAN data and discuss the economic impact analysis using IMPLAN. In Section 3, we present our results from the economic impact analysis. In Section 4, we present a qualitative analysis of the potential impacts of the circular economy (CE) and the sustainability of farmers' markets on the local economy using data obtained from focus groups. In the last section of the paper, we discuss the implications of this research and opportunities for future research.

### 1.1. Literature on Economic Impacts of Farmers' Markets and the Circular Economy
Economic Impacts of Farmers' Markets

The demand for farmers' markets has grown for the following reasons: the rise in consumer demand for fresh, locally grown produce; improvements in agricultural practices; and consumer interest in direct interaction with the growers [2]. Farmers' markets have been reported to provide economic benefits to producers, consumers, and local communities [3,4]. The economic impacts of farmers' markets have been studied and broken down in different ways. Hughes et al. [5] evaluated the direct and indirect economic impacts of farmers' markets on South Carolina's economy as well as the impact of the Certified South Carolina Grown Campaign, which emphasized patronage at farmers' markets. Farmers' markets have produced improvements in local economic development, enabled by having a site for local and small business incubation, creating an economic multiplier effect from the markets to other local businesses, and recirculating customer expenditure within the local community [6]. To assess these economic impacts, Sadler et al. [6] used farms and divided them into livestock and crop producers across three size categories, which resulted in six farm categories. Henneberry et al. [7] conducted a survey of Oklahoma farmers' market vendors. They estimated that the total gross farmers' market sales in the 2001 season was USD 3.3 million, with USD 7.8 million in direct and indirect effects on the economy of Oklahoma. Their analysis estimated that the USD 630,000 spent by consumers in other sectors led to a total statewide impact of USD 1.9 million, and 795 jobs directly generated by farmers' markets sustained an additional 1145 jobs in related activities. They estimated these economic impacts on Oklahoma's economy using an IMPLAN model. To supplement the IMPLAN model, they used survey data of total farmers' market gross sales, the number of people employed by farmers' markets, the annual average of farmers' market producers'/vendors' household income, and total farmers' market visitors' expenditures in other sectors. The empirical results showed that farmers' markets' activities are an important part of Oklahoma's economy because the farmers' markets generated total direct sales of USD 3.3 million with a total economic impact of almost USD 6 million. Survey data are often used in the measurement of economic impacts of farmers' markets, and Oberholtzer et al. [8] also used farmer surveys in their empirical analysis. Not only were important economic impacts revealed, but also that both farmer and market characteristics are important incentives for participating in farmer sales.

Hughes et al. [5] also estimated the economic impacts of farmers' markets in West Virginia using a survey of vendors at farmers' markets in 2005, including the value of total sales by local producers at farmers' markets. Their analysis estimated annual direct sales

(USD 1.725 million). As with other economic impact studies, they used an IMPLAN-based input-output model. Their estimates indicated 119 jobs generated (69 full-time-equivalent jobs) and USD 2.389 million in output, which included USD 1.48 million in gross state product (GSP). When they expanded the analysis to include the effect of direct revenue losses for primarily grocery stores, the economic impact was reduced to 82 jobs (43 full-time equivalent jobs) and USD 1.075 million in output, which included USD 0.653 million in gross state product (GSP). In their analysis, they provided detailed information on the categories of crops sold matched with sales levels, which were used to estimate the percentage distribution of sales of the different crop categories in IMPLAN.

In addition to economic impacts of farmers' markets, research has been done to examine the social impacts of farmers' markets. The research collected consumer demographics, utilization, satisfaction, and eating and physical activity behavior information from the consumers from farmers' markets in low income urban communities in East and South Los Angeles from April, 2007 through June, 2009 [9].

Rossi et al. [10] explored the local economic impacts of local compared with conventionally produced and marketed food in two regions in Missouri and one region in Nebraska. Their analysis determined that local food systems generated substantial increases in the value added to the local economies defined in the study. The analysis revealed that enhancements in local food markets improved local economic development. Differently stated, local sales resulted in greater regional economic impacts compared with traditional retail food markets. In addition, they found that local food operations allocate more of their expenditures to labor.

### 1.2. The Circular Economy and Sustainability

The circular economy (CE) is not a new paradigm in economics; it has been gaining momentum since the late 1970s. The authors in [11,12] attribute the introduction of CE to Pearce and Turner in their seminal work [13]. The circular economy describes how natural resources influence the economy by providing inputs for production and consumption and how waste from consumption is used. The circular economy aims to replace the linear production systems by enabling the production line to use waste to achieve greater local economy sustainability. In the food system, we need to change the food production and distribution systems, creating an efficient food chain through better use of resources. Achieving the latter efficiency will require moving toward a circular economy, which is true for cities because cities are expected to consume 80% of the food produced by 2050.

Before the conceptualization by Pearce et al. [13], Boulding's conceptual framework [14] described the Earth as a closed and circular system with little assimilation, and Stahel et al. [15] introduced certain features of the CE in an economic framework. They depicted the economy as a loop economy to describe industrial strategies for waste prevention, regional job creation, resource efficiency, and dematerialization of the economy. However, they did not formally define the CE. Stahel formally defined the closed-loop economy or the CE [16]. He described the impact of an economy as loops of resource savings, prevention of waste, creation of jobs, improving the role of innovation, and the development of a robust private sector. More importantly, he emphasized using selling instead of ownership of goods as a sustainable business model within a loop economy. The latter enables industries to profit without having to externalize costs and the risks associated with the production of waste.

### 1.3. Circular Economy and Food Production

In the past 100 years, agricultural systems have dramatically evolved, providing food to the increasing world's population and supporting regional economic development and rapid urban growth. These achievements have come at a cost, as the practices have been proven to not be sustainable [16]. The MacArthur Foundation and Geissdoerfer et al.'s framework [17] suggested three interrelated ambitions that businesses, governments, and

cities can use to allow food systems to be more sustainable. To achieve a CE for food production and distribution, they proposed the following:

- Using regenerative techniques to grow food and, where appropriate, locally;
- Improving food distribution, food waste, and waste management;
- Designing and marketing healthier food choices to consumers in terms of nutritional value and how the food is produced.

One of the main impacts of the CE in food production systems is to minimize food loss and food waste. Food loss and food waste result in economic loss, with financial losses of up to USD 1 trillion per year globally under the current food supply model [18,19]. The latter affects all actors in the supply chain, including consumers [20]. More importantly, our current food supply chain process wastes natural resources, e.g., soil and water. More importantly, these current agricultural practices create unnecessary pollution, leading to environmental pollution and degradation which is currently tied to a food production system that is linear [21].

As shown in Figure 1, the food production system adheres to a linear model because the resources needed to produce the food products start at the beginning and move linearly toward the end to the final food product. The final stage of the linear process shows the food being processed or consumed, which creates organic waste from the discarded food, its byproducts, or sewage. More important, in this linear food production process, less than 2% of the usable nutrients in the discarded food are returned to productive use [19–21].

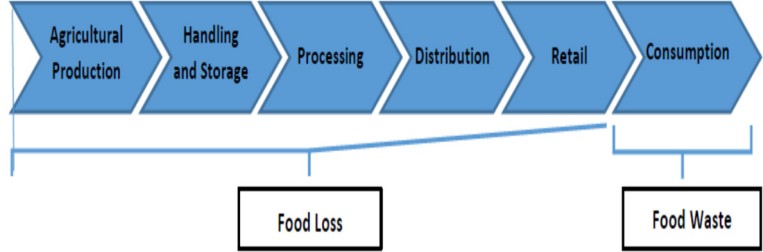

**Figure 1.** The Linear Food Production Process [22].

*1.4. Framers' Markets Impact on the Circular Economy*

Without a fundamental transformation of our entire food system, we will not be able to achieve a more sustainable environment, a healthy population, or a healthier planet. More importantly, sustainable changes to the environment can provide solutions to the changes in the climate. Adopting the tenets of the circular economy as a remedy to climate change may result in benefits for food security as well as improvements in water, forests, wetlands, pollution, and human health. Despite the positive economic contributions of farmers' markets, care must be exercised, especially the use of specific data as the unit of analysis of a study to ensure that the results from the economic impact analysis provide an accurate portrayal of the farmers' markets in a local or regional economy [22,23].

There is cognizance that the linear food production process needs to be revamped to minimize food loss and food waste, which are prevalent. A paradigm shift is needed to a circular economy from the current standard, linear economy. This paradigmatic shift will reduce waste, encourage the practice of more sustainable production methods, and integrate increased food production and purchase at the local level to reduce environmental degradation [23,24].

## 2. Materials and Methods
*Data Sources*

We used an input–output model, Impact Analysis for Planning (IMPLAN), in this analysis to understand the contribution of farmers' markets to the Washington, DC metropolitan economy. The IMPLAN model is used to determine how local changes as defined by the researcher affect a regional or a state economy. In this study, we supplanted the 2017 UDC

farmers' markets consumer survey data with the IMPLAN data to estimate the impacts of farmers' markets on the economy of the Washington, DC metropolitan area. Table 1 provides the profile of the study region.

**Table 1.** Description of the Washington, DC metropolitan area in 2017.

| Indicator | Value |
|---|---|
| Gross Regional Product (USD) | 580,779,309,002 |
| Total Personal Income (USD) | 463,617,884,800 |
| Total Employment | 4,848,299 |
| Number of Industries (NAICS) | 464 |
| Land Area (sq. miles) | 7733 |
| Counties in Study Area | 22 |
| Population | 7,023,227 |
| Total Households | 2,562,601 |
| Average Household Income (USD) | 180,917 |

Source: IMPLAN 2017, 22 Counties in Washington, DC Metropolitan Area. The twenty-two counties used in this study area are not homogenous, e.g., some are rural counties and some are urban, so heterogeneity exists in these counties. Table 1 presents the 22 counties total. The twenty-two counties are as follows: Maryland: Anne Arundel, Calvert, Carroll, Charles, Frederick, Howard, Montgomery, Prince George's, St. Mary's, and Washington; Virginia: Arlington, Clarke, Culpeper, Fairfax, Fauquier, King George, Loudoun, Prince William, Rappahannock, and Stafford; West Virginia: Jefferson; District of Columbia.

The total population of the study area was 7,023,227 in 2017, with a total of 2,562,601 households in 2017. The value of the gross regional product (GRP) or the gross domestic product of the study area was USD 580,779,309,002 in 2017 dollars with a total personal income of USD 463,617,884,800, also in 2017 dollars, supported by 4,848,299 employees in the study region. The average household income in the study area was USD 180,917 in 2017.

When using IMPLAN, the analysis requires specific data on local farmers' markets because of the potential differences between regional production functions and the national production function implicitly assumed by IMPLAN. There might be some differences in technology, adoption of technology, resource prices, and bias resulting from the aggregation of multiple industries in a single IMPLAN sector. If these differences exist, the accuracy of the results reported by IMPLAN will widely differ from reality. Consequently, the use of the primary data gathered by researchers in conjunction with the use of IMPLAN inevitably leads to higher accuracy of the results of the economic impact [22].

IMPLAN requires the use of input data, such as the use of production or expense data. However, in our study, we used consumer data as the inputs, which is the unique contribution of this study, because prior farmers' markets studies, as mentioned in the preceding section, used production or expense data. As a first step, we conducted a randomized survey in 2017. A sample of 767 members from Gfk KnowledgePanel (formerly Knowledge Networks), a probability-based web panel designed to be representative of the United States, was gathered. In total, 440 completed samples were delivered to the University of the District of Columbia (a completion rate of 68%). The survey results were obtained from farmers' markets shoppers no more than 20 miles from the DC city center, which included shoppers from Virginia and Maryland.

From the survey results, we extracted the following variables:

● The number of times attending farmers' markets;
● The amount spent at the farmers' markets;
● The commodities purchased at farmers' markets.

We used the latter data and prepared estimates of expenditure by commodity type. Table 2 provides the complete list of the commodities and dollar amount.

**Table 2.** List of commodities from Gfk KnowledgePanel survey.

| Commodity | Annualized Dollar Amount (USD) |
| --- | --- |
| Fruits | 3,547,585.62 |
| Vegetables | 4,069,976.72 |
| Nuts | 146,567.18 |
| Milk | 645,334.97 |
| Ice Cream | 658,611.54 |
| Cheese and Butter | 174,061.80 |
| Eggs | 880,238.87 |
| Meat | 1,646,696.94 |
| Honey | 59,368.71 |
| Fish and Seafood | 410,392.58 |
| Prepared Foods | 22,192.50 |
| Other Goods | 745,195.24 |

In the survey, there was a question concerning the number of times the person went to farmers' markets between May and November. Then, each of the expenditures was annualized and summed. These annualized total expenditures by each commodity were inputted into IMPLAN. The category Other Goods included fresh-cut flowers, beverages, and other commodities.

■ Economic Impacts and Multipliers

To assess the economic contributions of farmers' markets to the metropolitan area of Washington, DC's economy, three different estimates for direct, indirect, and induced effects were calculated using IMPLAN.

- Direct effects are the purchases made by the customers shopping at the farmers' markets.
- Indirect effects are the purchases of supplies and services that are provided to farmers' market producers and its vendors when the workers in the direct industry (the farmers markets) and those in the indirect industries (the supplying producers/vendors for the farmers' markets) convert their labor income into household spending.
- Household spending induces a third round of economic activity. These induced activities provide increased sales of all other businesses, e.g., retail stores in the area, because there is more spending in the metropolitan Washington, DC area resulting from the income generated by households from the direct and indirect activities from the farmers' markets.

IMPLAN also estimated the effects of these sales on the number of jobs in the study region; IMPLAN is an input–output (I-O) model including a matrix of several economic sectors in which the sectors along the horizontal axis represent the productive inputs to the industries on the vertical axis. More specifically, each cell of the matrix is linked to all of the other cells via some production function. Consequently, changing the values of goods supplied or demanded by any of the industries causes the model to change the matrix, which shows how that initial change affects all industries and how the supply inputs to or demand outputs from the industry are altered. For the analytical details of the I-O model, the reader should refer to [25].

■ Economic Impact Multipliers

From these impacts, we calculated the multipliers that show the effects of changes in the final demand of one industry on all other industries within our defined study. Depending on how the analysis is developed, multipliers can be estimated for a county, for groups of counties, or even an entire state. These economic multipliers were calculated from the IMPLAN model using the input data on consumer spending at the farmers' markets in the Washington, DC metropolitan area. IMPLAN was used to calculate the Type I and Type SAM multipliers, and these two multipliers provided different measures of total economic impacts.

The first multiplier estimated was the Type I multiplier. These multipliers only include the business-to-business purchases without the effects of household spending in the local area. The Type I multiplier is calculated as:

$$\text{Type I Multiplier} = \frac{\text{Direct Effects} + \text{Indirect Effects}}{\text{Direct Effects}} \tag{1}$$

We also estimated the Type SAM multiplier, which consists of the Type I multiplier and the inclusion of household spending. Type SAM is a commonly used and reported multiplier. The Type SAM multiplier is calculated as:

$$\text{Type SAM Multiplier} = \frac{\text{Direct Effects} + \text{Indirect Effects} + \text{Induced Effects}}{\text{Direct Effects}} \tag{2}$$

This paper presents the results from the economic impact analysis, including the Type I and Type SAM multipliers.

■ Adjustment of IMPLAN Estimates

Because we did not have the number of customers who shopped at farmers' markets in the DC metropolitan area, IMPLAN could have overestimated the farmers' markets' direct, indirect, and induced effects. To ensure our IMPLAN results were as close as possible to reality, we carefully reviewed the literature to identify the estimates of shoppers attending farmers' markets across the United States. After obtaining these estimates from the literature, we used these estimates and developed a methodology to adjust the direct, indirect, and induced effects produced by IMPLAN. The steps to prepare these estimates were as follows:

- We collected the percentage of shoppers attending farmers' markets across the U.S. and used an average of these estimates from the literature. We established the low-point participation (2.14%), high-point participation (4.58%), and mid-point participation (3.05%).
- We estimated the percentage of shoppers attending farmers' markets by estimating the regression of the number of farmers' markets in each region (Table 3) onto the percentage of shoppers in the region found in the literature. We inserted the number of farmers' markets in the DC metropolitan region, which was 154, into the regression equation to obtain an estimate for our study area.

**Table 3.** Calculation of the ratios of consumers and the population.

| Citation | | Farmers' Markets from Study | Consumers from Study | Population from Study | Ratio of Consumers and Population |
|---|---|---|---|---|---|
| [4] | Iowa | 189 | 135,000 | 2,942,000 | 4.589% |
| [6] | Flint, Michigan | 1 | 9197 | 450,000 | 2.044% |
| | London, Ontario | 10 | 7211 | 450,000 | 1.602% |
| [7] | Oklahoma | 29 | 42,000 | 3,751,351 | 1.120% |
| [9] | East and South LA | 25 | 1789 | 226,458 | 0.790% |

Because the above ratios are based on older data from the literature and the number of farmers' markets has increased throughout the United States since the publication of these studies, we adjusted these ratios by estimating the growth rates of farmers' markets. We adjusted these estimates by applying growth rates to the original ratios of consumers to the population.

We found updated estimates for the farmers' markets for Oklahoma, East and South Los Angeles, and Iowa for 2017 (the year of our IMPLAN results). The updated number of farmers' markets were obtained from local food directories: National Farmers' Market Directory, Agricultural Marketing Service https://www.ams.usda.gov/local-food-directories/farmersmarkets (accessed on 31 May 2020). However, we were not able to find updated estimates of the number of farmers' markets for Flint, Michigan and London, Ontario, Canada, so we assumed they remained the same.

- We calculated the growth rates from the originally published estimates to the updated estimates. The new estimates were:

  o   2.8637%, the average of the four regions without Ontario, Canada—the lower bound using the adjusted ratio;

  o   4.1256%, the predicted value for DC based on the estimation of a regression of the number of farmers' markets in the region onto the percentage of shoppers in the region;

  o   5.5113%, the ratio of Iowa of consumers to population—the upper bound used the adjusted ratio.

Table 4 summarizes how the adjusted ratios were calculated.

**Table 4.** Adjusted ratios of the number of farmers' markets [3] (column number in parentheses).

| Citation (1) | Region (2) | Number of Farmers' Markets (3) | Consumers from Study (4) | Population of Study (5) | Ratio of Consumers to Population (6) | Number of Farmers' Markets, 2017 (7) | Growth Rate of Farmers' Markets [1] (8) | Adjusted Ratios Using Growth Rates [2] (9) |
|---|---|---|---|---|---|---|---|---|
| [4] | Iowa | 189 | 135,000 | 2,942,000 | 4.59% | 227 | 0.2011 | 5.51% |
| [6] | Flint Michigan | 1 | 9197 | 450,000 | 2.04% | 1 | 1 | 2.04% |
| | Ontario, Canada | 10 | 7211 | 450,000 | 1.60% | 10 | 1 | 1.60% |
| [7] | Oklahoma | 29 | 42,000 | 3,751,351 | 1.12% | 74 | 1.5517 | 2.86% |
| [9] | East and South LA | 25 | 1789 | 226,458 | 0.79% | 33 | 0.32 | 1.04% |

[1] The growth rate of the number farmers' markets is between the number of farmers' markets, 2017 (7) and the number of the farmers' markets of the original study (3). [2] The adjusted ratios are the adjustments to the ratio of the consumers to population in column (6) using the growth rates in column (8). [3] After estimating these new ratios, we re-estimated the regression equations using the same approach as mentioned earlier in this section to obtain the range of estimates 2.61%, 4.13%, and 5.51%.

## 3. Results

Table 5 presents economic impacts of farmers' markets for our study region. The table summarizes these economic impacts based on the range of estimates calculated in the preceding table: 2.61%, 4.13%, and 5.51%.

As mentioned earlier, the direct, indirect, and induced effects on the output, employment, value-added, and labor income resulting from farmers' markets in the 22-county study region were estimated. All the values are expressed in year 2017 USD and reflect the total three impact components mentioned above. As expected, the direct effects accounted for the largest portion of the total economic impact in each impact category, or approximately 66% of the total output, 60% of the value-added, 83% of employment, and 44% of labor income. As for the indirect effects, it contributed to 20% of the output, 12% of the employment, 36% of the labor income, and 24% of the value-added. Finally, the induced effects accounted for 13% of the total output, 5% of the employment, 16% of the value-added, and 20% of the labor income. These economic impact estimates are for one year, not considering the continuing impact that may occur within the study region beyond one year.

As sector-specific multipliers differed by the IMPLAN sectors, we provide the aggregate multipliers for Type I and Type SAM for the output. The Type I multiplier was 1.31, whereas the Type SAM was 1.51. The Type I multiplier indicates that a USD 1 increase in income for a farmers' market yielded a USD 1.31 in personal income across the study region. For the Type SAM, recall that this multiplier includes the social accounting framework or the induced effects. The Type SAM multiplier indicates that a USD 1 increase in income for a farmers' market yielded a USD 1.51 in personal income across the study region.

**Table 5.** Summary of the results of the economic impacts.

| **Using 2.61%** | | | | |
| --- | --- | --- | --- | --- |
| Impact Type | Employment | Labor Income (in USD) | Value-Added (in USD) | Output (in USD) |
| Direct Effect | 354.63 | 2,455.513 | 7,375,973 | 15,436,000 |
| Indirect Effect | 51.91 | 2,015,725 | 2,915,333 | 4,751,434 |
| Induced Effect | 19.83 | 1,092,820 | 1,937,956 | 3,076,794 |
| Totals | 426.38 | 5,564,058 | 12,229,261 | 23,264,228 |
| **Using 4.13%** | | | | |
| Impact Type | Employment | Labor Income | Value Added | Output |
| Direct Effect | 560.72 | 3,882,513 | 11,662,455 | 24,406,495 |
| Indirect Effect | 82.08 | 3,187,145 | 4,609,552 | 7,512,688 |
| Induced Effect | 31.36 | 1,727,903 | 3,064,182 | 4,864,845 |
| Totals | 674.16 | 8,797,561 | 19,336,189 | 36,784,028 |
| **Using 5.51%** | | | | |
| Impact Type | Employment | Labor Income | Value-Added | Output |
| Direct Effect | 748.43 | 5,182,228 | 15,566,591 | 32,576,841 |
| Indirect Effect | 109.56 | 4,254,078 | 6,152,651 | 10,027,643 |
| Induced Effect | 41.86 | 2,306,337 | 4,089,951 | 6,493,406 |
| Totals | 899.85 | 11,742,643 | 25,809,192 | 49,097,890 |

Table 5 shows the economic contributions of farmers' markets in the study region, and the IMPLAN data used for the study region were from 2017. In this analysis, we used the IMPLAN regional purchase coefficients to estimate the share of farmers' market activity in the study region that would remain in the study region [26]. Then, IMPLAN was used to estimate regional trade flows using some form of a constrained gravity model that combines trade flow data with economic measures. The latter approach is an effective method to measuring trade flows across the designated study region.

We estimated the range of the value of farmers' market economic activity or the direct effect to be USD 15.4 to 32.5 million in 2017. Once we added the indirect and induced effects, the total transactions ranged from USD 23 to 49 million of additional economic activity, as shown in Table 5. Consequently, the latter transactions would have created 427 to 900 new jobs, with a range of 355 to 748 jobs created by the direct effect. More importantly, these jobs supported some USD 5.5 to 11.7 million in labor income in the study region, as shown in Table 5.

## 4. Discussion

### 4.1. Qualitative Data of the Possible Presence of a Circular Economy in DC Farmers' Markets

In the past few decades, consumers, researchers, and policymakers have been interested in sustainable food consumption, and farmers' markets have played a major role to promote sustainable agricultural production and consumption by consumers [27,28]. Farmers' markets promise to promote sustainability in local areas has attracted much attention. Even though we presented the economic impacts of farmers' markets on the Washington DC metropolitan area, we are interested in understanding the extent by which consumers were interested in promoting sustainability to the local economy via the circular economy of food production [29] as well as the attitudes towards sustainability of local farmers'

markets [30]. This section presents the discussion concerning the qualitative aspects of the possible presence of a circular economy (CE) in DC farmers' markets using focus groups.

### 4.2. Settings and Participants

The UDC IRB committee approved the human subject aspects of the proposed focus groups. Before the commencement of the focus groups, the focus group participants were provided a copy of a written informed consent form along with point of contact information of the principal investigator. These signed informed consent forms were not collected because it would have provided participant identifiers.

Four focus groups were conducted during the summer of 2019 at various locations throughout the Washington, DC area. These focus groups were part of a research program that explored consumer shopping patterns at farmer' markets. Two of the focus groups invited consumers who shopped at farmers' markets. The remaining two focus groups invited participants who were policymakers, policy analysts, economists, and other social scientists to obtain their perspectives on farmers' markets in Washington, DC.

### 4.3. Procedures

The focus group participants were recruited via a convenience sample from invitations to analyze policy and the dissemination of a recruitment flyer throughout the Washington, DC, area. Those who agreed to participate in the focus group were invited to participate at a specified location.

### 4.4. Data Collection

The moderator of the session provided a brief presentation at the start of each focus group to outline this research on farmers' markets. After the presentation, the moderator asked a series of questions regarding farmers' markets and asked follow-up questions, as needed. An assistant transcribed the notes from each focus group. Then, the notes were analyzed by themes.

### 4.5. Themes Related to a Possible Circular Economy

Theme 1: Improving Health by Having Access to Healthy Food Choices

Some of the participants contended that farmers' markets increase access to healthy food options, which may be limited in some neighborhoods. One participant described the importance of integrating an educational component at farmers' markets to improve healthier eating:

> I think part of the problem too is you don't know what you are cooking so then you don't know how to cook it. And that will address your body mass index (BMI) because that is an educational component related to fresh fruit and vegetables. I think it would be a good idea if you had an event where you had some type of cooking show. We could plan it out like seasons of Top Chef and actually sign up to be on the cooking show. The Chef working with two other people from the community and actually learn how to cook these things and compete additionally that would be a good way to teach the community.

However, another participant thought that having farmers' markets would not alleviate the problem of the lack of healthier foods in lower-income communities:

> No, in the bigger scheme of things, it puts a bandage on a huge problem. These two grocery stores here are supposed to feed 100,000 people for this Ward, so if you bring in a couple of farmers markets, it is not going to help the bigger problem because the grocery store has more to offer other than fruits and vegetables, things like meat.

A participant stressed the need to promote healthy food products at farmers' markets through the packaging of the healthier food choices:

> In low-income communities, everyone is busy, so we had discussed one time how to prepare the vegetables, cut them up, or make a package with a label on it showing people how to cook them so it can be quick and easy. Packaging it up to look friendly would be easier for the community, or we need to say this will cook in 7 min. Whether you are low-income or middle class, you are still spending money on food, so it is not like they don't have the money for food. It is just a lot of the times they just don't want to deal with it because it is not quick and easy for them.

A participant stressed the need to promote the good products at farmers' markets, so the consumers are willing to purchase them again.

> I am really hyping it up; you know its pesticide free, and there are the health benefits. I have a standard price, and when people start to walk away, I really want them to buy the food, and I do not want to take it home. Also, I want people to experience the difference in taste and the health benefits for them to come back. They do come back, but they do not want to pay more.

Another participant stressed the need for freshness, "One reason why I feel strongly about freshness is because of the evolution of the grow season produce. Different things coming in when they are in season gives validation of the farming."

Theme 2: Strategies in Developing Financially Sustainable Farmers' Markets in Low to Moderate-Income Communities

Some participants stated that there are certain methods to attract consumers to farmers' markets. They provided their reasons as follows:

- Availability of seasonal produce and quality and variety;
- Friendliness of the vendors, interacting with new people, and a sense of community;
- Seasonal fruits and vegetables;
- Convenience.

A participant provided some reasons why farmers' markets fail in lower- and moderate-income communities, "Low population density (foot traffic, proximity that could support the market), market hours, lack of community interest, engagement, lack of community partners, and food price comparable to preferences."

A participant stressed the need to have a strong outreach within the community to promote farmers' markets:

> I work in FreshFarm and our mission is to support farmers. We also need programs that help people access food. Farmers set their prices; perhaps it is a reasonable price based off the work they put into growing it. But it's not necessarily a price point that a lot of people can access, and then the solution that we turn to is like developing our programs. I think they are very confusing to explain to people. I practice whatever speech I get and practice how to say it; I just try to keep it as simple as possible. For farmers to participate in some of these things, they want to do so, but it's like another bureaucratic process to go through to be certified in some of these programs. Then, I think part of our problem as well is we are not in places where people are in most need, so we can't convey information about our programs most effectively at times.

A participant stressed the need to build partnerships to help develop farmers' markets in low- and moderate-income areas. Another participant stressed the need for a good site selection that would help promote farmers' markets:

> I think that there is an issue with site selection. Do we systematically have a consensus if the community wants the farmers' market, or is there a higher priority for an online delivery service or something? I just wonder sometime when we start markets, do we start them for different reasons? Like, sometimes

there is a property developer that wants to bring more people to this space, and we think a farmers' market will draw them there?

Another participant stated that farmers' markets must better integrate into the community and may help improve financial sustainability:

The reason me and my wife go to farmers' markets is for a sense of community. Perhaps the communities work a different way, being tied to the community and working with the churches, schools, or other mechanisms. It's a meet and greet, and I get to meet farmers I have never met before. Farmers' markets can't stand alone; they need to be a part of the community, and it's more than just selling.

A participant who is a policy analyst stated that accessibility to healthy foods in lower-income communities is possible:

It's good but not self-sustaining financially. If you look at the data for SNAP (Supplemental Nutrition Assistance Program) and the Produce Plus and you add it up, you can see that it is concentrated. Actually, FreshFarms told me that the redemption of SNAP benefits at Dupont Market exceed what they get out of Dupont. That means all the markets throughout the city on SNAP are taking those benefits and taking them to Dupont market because that is the best market. That means Dupont must subsidize the cost, so the benefits need to be spread throughout the other markets. The success of the Columbia Heights market and the Crossroads market is due to their partners in the community by developing a CSA (Community Supported Agriculture) share for the elderly residents who can't walk to the market.

Theme 3: Market Characteristics of Farmers' Markets

Some of the participants described some market characteristics that are necessary for farmers' markets:

- Diversity of the growers, be accepted in the community by the consumers;
- Hire from the community, involving people in the immediate area, and working with the growers;
- Self-empowerment.

## 5. Conclusions

The popularity of farmers' markets enabled to bring producers and consumers together. This interaction with the producers and consumers enabled producers to earn their profits resulting after selling their products, and the consumers can purchase fresh products directly from the producers. This interaction with the consumers and producers increases the number of socially and environmentally sustainable food systems. That is, an explanation in the growth of farmers' markets is attributed to the idea that the products sold at farmers' market products are more sustainable [31–33].

The number of farmers' markets in the Washington, DC metropolitan area has increased in the past decade which coincides with increasing of farmer's markets [30,31] but little research has documented the profile of these farmers' markets. There is no past research of the economic contributions of these farmers' markets to the regional economy. This paper addresses this research gap by summarizing the Washington, DC metropolitan area farmers' market profile using data collected from a survey conducted in 2017. The survey pertained to the shoppers who visited farmers' markets in the study region. After compiling the data from the survey, an IMPLAN-based input–output model was applied to trace the economic impacts of these farmers' markets. This research showed a positive economic impact on the region which is consistent with prior research on the economic impact of farmers' markets [3–8]. The important contribution of this research is the use of consumer data to supplant the IMPLAN-based social accounting matrices to evaluate the direct, indirect, and induced economic impacts of the farmers' markets in the Washington DC metropolitan area while prior research [3–8] used producer or production data to

evaluate the direct, indirect, and induced economic impacts of the farmers' markets on a state or a region.

We also qualitatively examined the circular economy as it pertains to farmers' markets because there is a stronger emphasis on the promotion of sustainability in local economies [17,23,25,34–37]. As a supplement to the economic impacts, we qualitatively examined the possible existence of a circular economy in farmers' markets. Based on the responses of the participants in the focus groups, we found some strong evidence of a circular economy. Future research may more closely examine the existence of a circular economy through the development of economic indicators or other empirical approaches. The latter would lead to improved sustainability in the local economy with spillover effects into the regional economy.

We suggest that a careful evaluation be conducted of the opportunity costs resulting from shoppers at farmers' markets who do not purchase similar items from retail food stores in future research. In this analysis, we used consumer data to estimate the sales at farmers' markets. To yield plausible estimates, we would need an accurate measure of gross farmers' market sales, which is crucial to determining economic impacts. Furthermore, shoppers may understate their purchases; therefore, the economic impacts in terms measured by sales, personal income, and jobs could be underestimated. Other direct-marketing outlets may exist in the study region, such as roadside stands, pick-your-own fruit/vegetable markets, and various types of agricultural cooperatives or food hubs that were not captured by the current estimates. We did not measure the impacts outside of farmers' markets. Thus, in future research, it may be useful to obtain a robust picture of the marketing of locally grown foods.

**Author Contributions:** Conceptualization, K.Z. and B.W.S.; methodology, K.Z. and B.W.S.; software, B.W.S.; validation, K.Z. and B.W.S.; formal analysis, B.W.S.; resources, E.C.H.; data curation, B.W.S. and E.C.H.; writing—original draft preparation, B.W.S.; writing—review and editing, B.W.S. and K.Z.; visualization, B.W.S. and K.Z.; supervision, K.Z.; project administration, K.Z.; funding acquisition, K.Z. All authors have read and agreed to the published version of the manuscript.

**Funding:** This research was funded by USDA/National Institute of Food and Agriculture (NIFA) block grant number 2015–2018.

**Institutional Review Board Statement:** The study was conducted according to the guidelines of the Declaration of Helsinki, and approved by the Institutional Review Board of University of the District of Columbia. The responses about farmers' markets were anonymous and no personal identifiable data were gathered.

**Informed Consent Statement:** Informed consent was obtained from all subjects involved in the focus groups of this study.

**Data Availability Statement:** Not applicable. The data from IMPLAN and the data from the Gfk KnowledgePanel are proprietary and not available for researchers.

**Conflicts of Interest:** The authors declare no conflict of interest.

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
