# Peer review of "Economic Impact Analysis of Farmers’ Markets in the Washington, DC Metropolitan Area: Evidence of a Circular Economy"

_sustainability, doi:10.3390/su13137333_

Round 1
Reviewer 1 Report
In my opinion, if the article is to be published, it needs a solid improvement. Besides, the applied research methods are too trivial

Reviewer 2 Report
The article is a highly skilled work. The authors are professionals. The processed statistics are analyzed properly. The authors showed the ability to think logically and operate with statistics and the results of their processing. However, there are moments that could be made much clearer in this article.
The content of the hypotheses adopted in the study can, of course, be guessed, but a clear statement of these hypotheses is not visible. That is, we are talking about the need for a clearer formulation, such as: "the main hypothesis adopted in this study is that… Auxiliary hypotheses are…". At the same time, it would be good to show in the conclusions whether these hypotheses were verified during the study, and whether they came true.
The relationship between farmers markets and sustainability in the article is in some way traced by economic indicators of impact. But, if we correctly understand the essence of the circular model of the economy, the materials and waste used in it again become raw materials for the economy. Actually, I did not observe this aspect in the article. So, I have not seen the influence of farmers' markets on the formation of a circular type of economy. That is, obviously, there is a need to pay more attention to how farmer markets affect the formation of a circular type of economy.
Reviewer 3 Report
The authors have tried to bring a serious work with this paper. There are a few imperfections those must be improved:
In the introduction, originality must be justified in the context of previous studies. The author should update the manuscript with appropriate and relevant research questions. The importance of research does not arise. The authors only give the results without a corresponding explanation. It is better to do some data analysis to make the conclusion more believable. Further, what is the major contribution of this study? Try to illustrate them in the manuscript. The conclusions must be improved, clearly indicating the main conclusion, and relating it to the literature supporting the paper ("according to (...), or" unlike (...) "). Also, conclusions should be rewritten to understand the importance of research. Are there any further research streams?
Reviewer 4 Report
I read carefully the paper entitled " Economic Impact Analysis of Farmers Markets in the Washington, DC Metropolitan Area: Evidence of a Circular Economy".
The topic of the research is very important.
We have discovered many scientifically valuable elements.
I recommend the following to the authors to better identify the elements of their own scientific contribution.
In particular, to specify which is the part through which the paper brings superior elements in relation to other researchers.
I consider the Introduction to be long and too fragmented.
I recommend that there be a separate Introduction and a separate Literature Review.
The writing of the paper and the graphics do not follow the rules of the journal (Template).
The bibliography should be extended with some papers published in prestigious WoS indexed journals (2020-2021).
Authors can discover the errors themselves (quite a lot) if they reread the article.
Round 2
Reviewer 1 Report
I accept the authors' explanations
Author Response
Point: no comments. The reviewer was satisfied with our responses
Response: Thank you for taking the time to review our paper and to provide valuable feedback. Your feedback improved the paper.
Reviewer 3 Report
The authors responded to the comments.
Author Response
Point 1: The reviewer was satisfied with our comments. Just changes to English.
Response: We had a professional proofreader carefully review the paper to ensure that the English language was appropriate. Thank you for taking the time to review our paper and to provide valuable feedback. Your feedback helped to improve the paper.
Reviewer 4 Report
Thank you very much for giving me the opportunity to review this article. I find that the authors have made some improvements.
However, I noticed some weaknesses:
- Image quality is very poor (Figure 1. The linear food production process)
- The font size in Table 1 is not adequate (it is too large) - Some editorial negligence appears (Line 305)
- Editing the Bibliography does not follow the rules of the journal
-
The bibliography should be extended with some papers published in prestigious WoS indexed journals (2020-2021).
Author Response
Thank you for taking the time to review our paper and to provide valuable feedback. Your feedback improved the paper. See below our responses to your comments:
Thank you very much for giving me the opportunity to review this article. I find that the authors have made some improvements.
However, I noticed some weaknesses:
Point 1 Image quality is very poor (Figure 1. The linear food production process)
Response: After some additional work, we redesigned Figure 1. Though we modified the figure from a source, we also cited the source as [35].
Point 2 The font size in Table 1 is not adequate (it is too large) - Some editorial negligence appears (Line 305).
Response: We corrected these issues. Thank you for pointing out this issue and our editorial negligence.
Point 3 Editing the Bibliography does not follow the rules of the journal
Response: We edited the Bibliography to make sure it follows the rules of the journal.
Point 4 The bibliography should be extended with some papers published in prestigious WoS indexed journals (2020-2021).
Response: We looked closely at the Web of Science (WoS) for articles in 2020-2021. We were not able to locate articles from 2020-2021 but found some articles from 2019 which are pertinent. The articles are listed in the Bibliography as [33-34].